# Mold Fungal Resistance of Loose-Fill Thermal Insulation Materials Based on Processed Wheat Straw, Corn Stalk and Reed

**DOI:** 10.3390/polym16040562

**Published:** 2024-02-19

**Authors:** Ramunas Tupciauskas, Zigmunds Orlovskis, Karlis Trevors Blums, Janis Liepins, Andris Berzins, Gunars Pavlovics, Martins Andzs

**Affiliations:** 1Laboratory of Biorefinery, Latvian State Institute of Wood Chemistry, Dzerbenes 27, 1006 Riga, Latvia; andris.berzins@kki.lv (A.B.); pavlovichs@inbox.lv (G.P.); martins.andzs@kki.lv (M.A.); 2Faculty of Biology, University of Latvia, Jelgavas 1, 1004 Riga, Latvia; zigmunds.orlovskis@lu.lv (Z.O.); ktblums@gmail.com (K.T.B.); janis.liepins@lu.lv (J.L.); 3Latvian Biomedical Research and Study Centre, Ratsupites 1 k-1, 1067 Riga, Latvia; 4Faculty of Forest and Environmental Sciences, Latvia University of Life Sciences and Technologies, Akademijas 11, 3001 Jelgava, Latvia

**Keywords:** wheat straw, reed, corn stalk, steam-exploded pulp, thermo-mechanical pulp, lignocellulosic biomass, loose-fill thermal insulation materials, mold fungi resistance

## Abstract

The present study evaluates the mold fungal resistance of newly developed loose-fill thermal insulation materials made of wheat straw, corn stalk and water reed. Three distinct techniques for the processing of raw materials were used: mechanical crushing (Raw, ≤20 mm), thermo-mechanical pulping (TMP) with 4% NaOH and steam explosion pulping (SEP). An admixture of boric acid (8%) and tetraborate (7%) was applied to all processed substrates due to their anti-fungal properties. The fourth sample group was prepared from SEP substrates without added fungicide (SEP*) as control. Samples from all treatments were separately inoculated by five different fungal species and incubated in darkness for 28 days at 28 °C and RH > 90%. The highest resistance to the colonization of mold fungi was achieved by TMP and SEP processing, coupled with the addition of boric acid and tetraborate, where molds infested only around 35% to 40% of the inoculated sample area. The lowest mold fungi resistance was detected for the Raw and SEP* samples, each ~75%; they were affected by rich amount of accessible nutrients, suggesting that boric acid and tetraborate additives alone did not prevent mold fungal growth as effectively as in combination with TMP and SEP treatments. Together, the achieved fungal colonization scores after combined fungicide and pulping treatments are very promising for the application of tested renewable materials in the future development of thermal insulation products.

## 1. Introduction

Thermal insulation materials are very important building construction materials that reduce the energy consumption of a building while simultaneously providing indoor comfort. When properly installed, thermal insulation materials significantly reduce greenhouse gas emissions including carbon dioxide (CO_2_) [1]. Therefore, the development of new eco-effective thermal insulation materials is essential to comply with the climate change policy first set by the United Nations in the Paris Agreement [2] and approved by the European Commission in the Green Deal [3], as well as to offset the rising costs of energy resources. In the context of the Green Deal, the development of renewable lignocellulosic biomass (LCB)-based materials from agricultural residues is particularly desirable because they contribute to a “green building” with nearly zero energy [4]. Moreover, it was declared that the production of LCB-based materials has many of advantages such as low environmental impact, less energy consumption, low cost, low density, scalability, biodegradability and good insulation properties [5]. A lot of studies considered diversifying the residual biomass pretreatment, including using hydrothermal, microwave, enzymatic and fungal techniques, for the development of sustainable building materials, including insulation boards, fiberboards, particleboards and wood–plastic composites [6,7,8,9]. However, still the share of LCB-based materials in insulation applications reaches only 10%, from which the largest share is virgin/recycled wood fibers and cellulose (recycled paper) [10]. This signifies the potential to develop and offer new products for the market.

While LCB-based materials used in thermal insulation are in demand because of their environmental friendliness, they are hygroscopic and therefore sensitive to colonization by microorganisms such as fungi and bacteria, challenging the longevity and favorable thermal properties of these materials. Depending on environmental conditions and installation method, the produced LCB-based insulation can be colonized by airborne fungal spores [11]. Moreover, the fungi-infected insulation impacts indoor air quality, which can harm the health of inhabitants [12]. Therefore, the durability of LCB-based thermal insulation materials also includes biological resistance which is dependent on temperature and humidity variations but is largely poorly investigated, as reviewed by Schritt and Pleissner [10]. The water-related thermal properties of LCB-based materials are also of high importance to be investigated [5] because varying humidity impacts not only biodeterioration but also the thermal properties [13,14]. Some natural mold resistance associated with the chemical composition of different wood species containing antifungal compounds was revealed on wood flour [15] and wood–plastic composites [16]. The microbial quality (mold and bacteria) of industrial crops (flax, hemp, straw) which potentially could be used as raw materials for insulation is affected by atmospheric conditions during growing season and they always present a wide diversity of molds [17]. An investigated rigid insulation panel from rape straw and hemp shives without added fungicides showed very different mold fungi growth rates, in general reaching over 25% from the area for hemp aggregates and no visible growth for rape aggregate [18].

Therefore, preservation treatment of LCB raw materials is necessary to prevent or decrease the mold growth on the end product. Boric acid and borates are commonly used as biocides in commercial cellulosic thermal insulation [19,20]. Cellulose insulation treated with sodium polyborate showed good results precluding fungi growth for at least 124 days at high temperatures and relative humidity [12]. The production of hemp fibers by steam explosion pulping was found to be a good process capable of decreasing fungal contamination, which is unfavorable in insulation materials [21]. None or marginal mold growth was detected on thermal insulation boards from corn pith and sodium alginate with 8% boric acid added [20]. The successful antifungal activity of silver nanoparticles was demonstrated on gypsum drywall and was suggested for a building material for the effective protection of indoor environments from mold development [22]. The effect of different solvents like cold/hot water, benzene-ethanol, ethanol-ether, NaOH and HCl was investigated on mold development on bamboo timber, revealing the best resistance to mold growth by adding 1% of HCl [23]. Surface treatment with castor oil-based polyurethane resin of particleboards from sugarcane and eucalyptus wood decreased the percentage of the mold colonization area even after 12 months of natural exposure [24]. Wood fiber insulation boards containing different mixtures of spruce and hardwoods produced in a dry process with PMDI adhesives and different additives demonstrated sufficient mold fungi resistance with a surface growth rate <50%. However, the results significantly varied depending on the test method [25]. Impregnation of bleached chemi thermo-mechanical pulp by hydrophobic betulin containing extractives from birch wood outer bark resulted in improved water- and fungal-resistance properties [26].

The studies reviewed above highlight the importance for further research into mold fungi’s effect on existing and newly developed LCB-based thermal insulation products. Therefore, our study continues the research of new thermal insulation materials [27,28] from locally sourced and annually harvested LCB such as wheat straw, water reed and corn stalk, and provides the results of mold fungi resistance. The research is significant due to the fact that the selected raw materials are widely available around the world and the obtained results could be useful for the development of and increase in LCB-based thermal insulation.

## 2. Materials and Methods

### 2.1. Raw LCB

Wheat straw (*Triticum aestivum*)—WS (grain extracted from Limbaži district, Latvia), water reeds (*Phragmites australis*, whole plant harvested in winter from Puzes Lake, Ventspils district, Latvia) and corn (*Zea mays*) stalks (fresh, ear/grain extracted from the farm “Pauri”, Blome, Latvia) were used in the study as locally grown raw materials. The delivered raw materials were chopped in a knife mill (CM4000, LAARMANN, Roermond, The Netherlands) to pass a sieve with openings of Ø 30 mm. The chopped LCB materials were used for further processing.

### 2.2. Processing of Raw LCB

#### 2.2.1. Steam Explosion Pulping (SEP)

Based on the previous studies [27,28], the chopped raw LCB was moisturized up to 80% moisture content by immersing it in water for 24 h. After, the LCB was drained and separately treated in a home-made SE device of original construction with a 0.5 L batch reactor at constant conditions: temperature of 230 °C, residence time of 30 s and maintaining a pressure of 30 bar. After, the wet SEP was collected and manually squeezed in a juice-like press to remove the liquid fraction.

#### 2.2.2. Thermo-Mechanical Pulping (TMP)

TMP of chopped and soda-treated LCB was performed in a single-disc refiner Regmed MD-300 (Osasco, Brazil) at constant 1450 rpm. The soda treatment was performed by adding 4% of NaOH based on dry LCB weight and cooking in water at the proportion 1:28 for 30 min. After, the soda-treated LCB was drained and subjected to the refiner fulfilled with water (20 °C). The duration of TMP process was constant (10 min) for all samples achieving a gap of 0.25 mm between universal plates. The resulting fiber solution was drained through a 2 mm sieve and manually squeezed in a juice-like press.

#### 2.2.3. Mechanical Foaming of Processed LCB

The obtained SEP and TMP materials were mechanically foamed by a self-made device through a system of two rotating cylinders (900 rpm) coupled with stainless steel wires, as described in [29]. The procedure was performed at least 3× to separate and homogenize the obtained fiber mass making it fluffy and suitable for application as loose-fill thermal insulation material.

#### 2.2.4. Admixture of Fire Retarder and Fungicide

To prevent the investigated materials from fire and biological effects, 8% of boric acid (H_3_BO_3_, CAS: 10043-35-3; Chempur, Piekary Śląskie, Poland) and 7% of di-Sodium tetraborate decahydrate (Na_2_B_4_O_7_·10H_2_O, CAS: 1303-96-4; Chempur, Piekary Śląskie, Poland) were added based on dry weight of LCB substrate. The substances first were solubilized in the hot (90 °C) water at the proportion 1:3 and then sprayed on the foamed LCB samples.

The control samples of unprocessed raw materials were prepared as well by chopping in the knife mill to pass a sieve of Ø 20 mm and addition of the above-mentioned substances. A control sample of SEP without admixture of fire retardant and fungicide substances was prepared additionally to detect the effect of SE treatment alone on mold fungi growth.

All the prepared LCB samples were conditioned prior to mold growth test in a chamber under controlled conditions (temperature 20 ± 2 °C and relative humidity 60 ± 5%) until equilibrium moisture content, which was 9.8 ± 0.4% for raw, 7.8 ± 0.1% for SEP and 10.3 ± 0.2% for the TMP samples, respectively.

### 2.3. Mold Growth Tests

Mold growth tests were performed following the procedure described in Annex F of [30] with fungal inoculates obtained from DSMZ—German Collection of Microorganisms and Cell Cultures GmbH. The following inoculates were used: A—*Trichoderma viride* strain DSM 1963 (synonym ATCC 9645), B—*Chaetomium globulosum* strain DSM 1962 (synomym ATCC 6205), C—*Paecilomyces variotii* strain DSM 1961 (synonym ATCC 18502), D—*Talaromyces pinophilus* (*Penicillium pinophilum*) strain DSMZ 1944 (synonym ATCC 36839), E—*Aspergillus niger strain* DSM 1957 (synonym ATCC 6275). All fungal isolates were grown on 0.5× PDA (Potato dextrose agar) for 14 days at 25 °C (in dark) until sporulation. Spores were collected by applying 10 mL sterile water onto the mycelial surface in each Petri dish (Ø 120 mm) and disrupting the mycelium with a sterile inoculation loop to resuspend the spores in the water. Spore suspension was passed through a column containing sterile cotton to remove fungal hyphae. Spores were counted using an automated cell counter LunaFX7 (Logos biosystems, Republic of Korea). In total, 25 mL of the uncompressed loose-fill LCB test specimens were evenly placed onto filter paper (Watman, pre-wetted with 4 mL sterile water) inside a Petri dish (Ø 90 mm).

Four pine (*Pinus sylvestris* L.) sapwood specimens (30 × 30 × 5 mm) per Petri dish were used as a control sample to verify the spore growth of each mold fungi. The delivered control specimens of pine sapwood contained different surfaces; these were characterized as tangential and radial as shown in Figure 1. Therefore, the selected fungi were inoculated on both wood sample surfaces.

Each specimen was spray-inoculated with 10^6^ spores (in 4 mL water) of a single fungal strain and Petri dishes sealed with Parafilm. Additionally, to the fungi-inoculated LCB samples, sterile water was used on each processed LCB material as control for any possible microbial growth originating from the processed test materials themselves. Four replicates were prepared for each combination of material type and fungal species and incubated for 28 days at 28 °C (in dark), >90% RH.

### 2.4. Evaluation of Mold Fungal Growth

All loose-fill LCB and wood control sample specimens (Table 1) were inspected under stereomicroscope (Leica MZ9_5_, Germany) for qualitative scoring using a relative scale from 0 to 5 adopted from [31]. Representative images were taken for each specimen. The mold growth rate of each test specimen was evaluated using the scale ranking as follows:0—no detectable growth;1—small growth with ~20% colonization;2—sparse growth with ~40% colonization;3—moderate growth with ~60% colonization;4—heavy growth with ~80% colonization;5—very heavy colonization across the entire material surface (~100%).

The factors of the influence on the mean values accomplished with standard variation of the evaluated sample’s mold fungi growth rate were analyzed by one-way ANOVA at the significance level α = 0.05 [32].

## 3. Results

### 3.1. Mold Fungal Colonization on Wood Controls

The results of the mold fungi growth on the wood control samples are summarized in Table 2 and supplemented by a microscopical surface view in Figure 2.

As can be seen from Figure 2, the mold fungi growth on the wood controls is significantly different depending on the fungi species and even wood surface. The growth of fungi A—*Trichoderma viride* and C—*Paecilomyces variotii* was observed only on the radial surface of wood specimens, with the fluffy hyphae covering up to 60% of it (Figure 2a,c). The growth of fungi B—*Chaetomium globulosum* was rated with 1, meaning that it covered the control specimens’ surface up to 20% by the small black points evenly on both tangential and radial surfaces (Figure 2b). The growth of fungi D—*Penicillium pinophilum* was not observed on both wood surfaces (Figure 2d), similar to the control samples incubated without fungi inoculation. Only the growth of fungi E—*Aspergillus niger* was detected on both wood surfaces, covering it up to 60% by obvious black points (Figure 2e).

### 3.2. Mold Fungal Colonization on Loose-Fill LCB

#### 3.2.1. Fungal Colonization on Wheat Straw LCB Samples

The results of the mold fungal colonization on loose-fill wheat straw LCB samples depending on fungi species are summarized in Figure 3 and supplemented by the surface views in Appendix A. As can be seen, all WS samples were affected by fungus growth rated on average from 1.3 ± 0.5 (WS-SE with fungi C—*Paecilomyces variotii*) to 4.0 ± 0.0 (WS-raw with fungus A–D). There was no detected significant difference of fungal colonization between all WS-raw samples infected with individual fungus A–E (Figure 4a and Appendix A), except for the control sample (H2O), which demonstrated a moderate colonization score of 3.4 ± 0.3 (Figure 3).

The fungal colonization of the WS-TMP samples varies on average in a range of 1.6–2.8 (small to moderate, Figure 3 and Appendix A); however, due to the high standard variation, the difference between the samples was calculated as insignificant (*p*-value 0.698). The microscopical surface view of the sample affected by fungi B—*Chaetomium globulosum* with developed sparse hypha is shown in Figure 4b and Appendix A.

The fungal colonization of the WS-SE samples (Appendix A) varies in an average range of 1.3–3.3 (small to moderate) with calculated significant difference between the samples (Figure 3). There is no significant difference between the fungal colonization scores of the samples infected with A and D fungus and between fungus B, C, E and control, respectively. This observation means that the fungal colonization score of the WS-SE samples depends on the individual fungi, in spite of fungicide presence. In turn, the fungal colonization score of the WS-SE* samples varies in average range of 2.8–4.0 (moderate to heavy) indicating significant influence of fungicide absence. However, the only *T. viride* demonstrates insignificant colonization difference between the SEP samples with and without fungicide addition, respectively (Appendix A). The last observation fits the previous tendency that the colonization score of the WS-SE samples depends on individual fungi in spite of fungicide presence, which can be seen in Figure 4c,d, Appendix A.

Summarizing the mold fungal colonization scores on WS samples, it could be said that these depend on individual fungi, the sample processing and fungicide application. As well, the used fungi are viable on all WS samples independent of the processing, including LCB control samples (H2O) without added fungi (Figure 3).

#### 3.2.2. Mold Fungal Colonization on Corn Stalk LCB Samples

The results of the mold fungal colonization on loose-fill corn stalk LCB samples depending on fungi species are summarized in Figure 5 and supplemented by the surface views in Appendix A. As in the case of the WS samples, all of the corn samples were also affected by fungus growth rated on average from 0.6 ± 0.3 (Corn-TMP, H2O) to 4.5 ± 0.4 (Corn-SE* with fungus D). If we compare with WS samples, the fungal colonization was higher for the Corn-raw and Corn-SE* samples, but lower for Corn-TMP and Corn-SE samples (Figure 3 and Figure 5). This observation could be associated with the specific structural difference of LCB species and their susceptibility to mold fungi depending on the processing.

The fungal colonization of Corn-raw samples depending on fungi species varies insignificantly in the range of 4.0–4.4, which is characterized as heavy (Figure 6a and Appendix A). The fungal colonization of Corn-TMP samples depending on fungi species varies in the range of 0.6–2.4, which is small to sparse (Figure 6b and Appendix A), and the variation is significant, which is the different resistance level of the material to individual mold fungi. The best mold fungi resistance is demonstrated by the control Corn-TMP sample (0.6 ± 0.3), but the worst resistance is detected with inoculated fungi E—*Aspergillus niger* (2.4 ± 0.5).

The fungal colonization of the Corn-SE samples is rated in the range of 1.0–3.8, which is small to heavy (Figure 6c and Appendix A), and the variation is significant. Small colonization is detected on the samples inoculated with fungi D—*Penicillium pinophilum* (Appendix A) and E—*Aspergillus niger* (Appendix A) and on the control sample (Appendix A). In turn, heavy growth is detected on the sample inoculated by the fungus A—*Trichoderma viride* (Appendix A). The fungal colonization of Corn-SE* samples depending on fungi species varies insignificantly in range of 4.1–4.5, which is characterized as heavy to very heavy (Figure 6d and Appendix A). This means that the Corn-SE* material, similar to Corn-raw, is very susceptible to all mold fungi species, including just the conditions of a humid environment which was tested by the control sample.

#### 3.2.3. Mold Fungal Colonization on Water Reed LCB Samples

The results of the mold fungal colonization on water reed LCB samples depending on fungi species are summarized in Figure 7 and supplemented by the surface views in Appendix A. The fungal colonization of all Reed-raw samples varies on average in a range of 2.3–4.5, which is characterized as moderate to very heavy (Figure 8a and Appendix A), and the difference between the samples is significant. Moderate growth (2.3 ± 0.5) is observed by fungi B—*Chaetomium globulosum* (Appendix A), but was very heavy (4.5 ± 0.0) on the control sample without inoculated fungi (Appendix A).

The fungal colonization of all Reed-TMP samples varies in an average range of 1.3–2.0, which is characterized as small to sparse (Appendix A). The ANOVA indicated insignificant differences between the samples spored with different fungi. The lowest fungal colonization score (1.3 ± 0.3) is observed on the control Reed-TMP sample (Figure 8b and Appendix A), which is different if compared with the Reed-raw sample (Figure 7).

The fungal colonization of all Reed-SE samples varies significantly in a range of 1.3–3.6, which is characterized as small to heavy (Figure 8c and Appendix A). The lowest fungal colonization score (1.3 ± 0.9) is observed by fungi B—*Chaetomium globulosum* (Appendix A), but heavy growth (3.6 ± 0.5) is observed by fungi A—*Trichoderma viride* (Appendix A). The fungal colonization of SEP samples without added fungicide (Reed-SE*) demonstrates a lower resistance varying in a range of 3.5–4.1, characterizing heavy growth (Figure 8d and Appendix A) depending on fungi species. Despite the small range of fungi growth, the ANOVA indicated significantly lower results for the Reed-SE* samples inoculated by fungus D—*Penicillium pinophilum* (3.5 ± 0.4) and E—*Aspergillus niger* (3.5 ± 0.0), Figure 7.

## 4. Discussion

In general, the wood control samples demonstrate a higher resistance to mold fungi when compared to the LCB samples (Table 2, Figure 3, Figure 5 and Figure 7), particularly taking into account that the wood controls were not pretreated by fungicide. The different intensity of mold fungi growth regarding the radial and tangential surfaces could be explained by a higher amount of soluble sugars found on the sapwood surface, as reported by [33]. From another point of view, the loose-fill LCB structure is more accessible to mold fungi than the surface of rigid wood. Another difference between the samples that could be considered is the moisture content that, possibly, increased faster in the LCB samples than in the wood due to the high humidity of the test. Moreover, the achieved equilibrium moisture content even at the same conditions depends on the LCB species. For example, the equilibrium moisture content of wheat and corn grains at air conditions of 28 °C and RH 90% achieves 16.7% and 18.9%, respectively, [34] indicating that the fungi development between LCB species could be different. The mold fungi growth rate of wheat straw–polypropylene composites was evaluated as 1 after one week exposure and after four weeks it reach 4, covering >60% of the composite surface [35].

The detected mold fungi growth at small to very heavy levels of all control LCB samples (H2O) could be explained by the presence of airborne mold fungi. As was reported, the airborne mold samples were consistent with bulk cellulosic insulation sample analyses relative to the mold types present [11]. Regarding the average values of mold fungal colonization of the control LCB samples, in general, they were at lower levels compared to the fungi-inoculated LCB samples (Figure 3, Figure 5 and Figure 7). Sufficient resistance to mold growth dependent on the raw LCB species showed that the TMP (1.4 ± 1.1) and SEP (1.4 ± 0.9) samples were the most appropriate processing techniques, combined with fungicide addition.

The highest intensity of mold fungi growth dependent on fungi species was observed for the unprocessed samples, in spite of the fungicide treatment. From them, the lowest mold fungi resistance was shown by corn samples, 4.1 ± 0.1 (Figure 5), followed by wheat straw, 3.9 ± 0.3 (Figure 3) and reed samples, 3.1 ± 0.8 (Figure 7). The difference between the samples’ average values was significant. That phenomena could be explained by the large amount of soluble sugars, providing nutrients for mold fungi [31]. The negative effect of higher sugar contents against molds was shown on wood–plastic composites made of different wood types [16]. The soluble sugars of raw LCB used in this study with the most potential for mold is the content of hemicelluloses, which varied in range of 24–30% [29]. The highest resistance to fungi growth of reed raw LCB could be associated with the natural plant growing conditions in water. However, as was reported by Malheiro et al. [36], the mold growth intensity of neat giant reed was detected between 4 and 6, covering 50–80% of the sample area, which fits our results. The high intensity of mold fungi growth on the unprocessed LCB samples could also be attributed to a low content of chemical components responsible for mold resistance. As antifungal components studied by Feng et al. [15], only minor contents of sitosterol and palmitic acid were found in wheat straw [37], palmitic acid and boron in common reed [38] and several phenolics in corn stalk [39]. Presumably for that reason, even the used fungicide content in raw LCB samples is not enough to prevent mold growth at such severe conditions in acceptable levels (<3).

The lowest fungal growth between the tested LCB samples was detected on all TMP samples (1.8 ± 0.8), which is acceptable for product use. The difference between the LCB species dependent on fungi species was found to be significant (*p*-value 0.017). It is expected that the obtained results were influenced by the development process, which included cooking with NaOH and then defibration in water medium. It was shown that bamboo timber treated with 1% of NaOH resisted *Trichoderma viride* and *Penicillium citrinum* efficiently; however, the fungi resistance against *Aspergillus niger* was lower [23]. The insulation panel of alkaline-treated and thermally compressed rape straw showed no visible mold fungi growth, demonstrating perfect fungi resistance [18].

The effect of applied fungicide on loose-fill LCB in the framework of this study is obvious; however, it is not enough in all cases. For example, comparing all the SEP samples dependent on raw LCB and fungi species, the fungal colonization scores of the fungicide-sprayed samples is ~2× lower (2.0 ± 1.1) than those without fungicide application (3.8 ± 0.7). The positive effect of steam explosion treatment on mold growth reduction on hemp fibers was also approved by Nykter et al. [21]. If we compare the fungal colonization of all raw (3.7 ± 0.7) and SEP (2.0 ± 1.1) samples, the SE effect is significant, which is associated with a reduced amount of hemicelluloses in the SEP samples, resulting in a lower wettability [40]. However, it does not work without fungicide. For that reason, additional water rinsing especially combined with NaOH addition after the SE process may be suggested to release the residual hydrophilic hemicelluloses on the fiber surfaces. Possibly, this would result in a decrease in fungi growth, indicating that a fungicide would not be necessary, taking into account that insulation materials are anticipated to be located inside a construction under conditions that are less harsh than in the performed mold fungi test. For, example, it was observed on chemically untreated wet-sprayed cellulose insulation that mold fungi growth significantly correlates with sample moisture and RH, and these units reduce the fungi growth from 75% to 25% [12].

It should be noted that the fungal colonization of used LCB samples was different depending on the fungi species and the LCB processing. Different fungal activity at varying conditions depending on individual fungi was demonstrated by Li and Wadso [41], concluding the effect of optimal fungi colonization conditions. The processing effect of LCB could be evaluated by the growth intensity of fungi A—*Trichoderma viride*, which was detected as moderate for the Reed-raw sample (3.5 + 0.6), as seen in Figure 7; the same intensity of the fungi was detected after SE with fungicide (Reed-SEP, 3.6 ± 0.5) and increased on the sample without fungicide (Reed-SEP* 4.0 + 0.0). However, the fungi A growth intensity decreased significantly after the TMP process (Reed-TMP, 1.8 ± 1.0). If we compare the detected average growth intensity of the used fungi species dependent on LCB and processing, fungi A—*Trichoderma viride* (3.3 ± 1.2) shows the highest intensity shows, but (2.7 ± 1.1)—fungi E—*Aspergillus niger* shows the lowest.

The same variation tendency depending on LCB processing was observed also on the control (H2O) samples, of which the average mold growth intensity was calculated as 2.6 ± 1.5. The best fungal growth resistance between these samples was observed for the Corn-TMP (1.5 ± 0.6) and Corn-SEP (1.6 ± 1.1) samples (Figure 5), demonstrating the effective influence of processing on the end products, which is suggested for insulation applications. However, the fungal growth resistance of the sample Corn-SE* (without fungicide) on average was rated by 4.4 ± 0.5 with a fungi-covered area of about 87%, proving the influence and necessity of fungicide.

The performed study provides knowledge regarding the mold fungal resistance of processed LCB for applications in thermal insulation. Since the selected LCB species are widely available around the world, the results of the study are of high importance. Taking into account that TMP and SEP processing resulted in the best results for all used species, the study will be continued by long-term testing of mold fungi in real construction; as well, we will investigate other important properties like settlement, water vapor diffusion, reaction to fire and volatile organic compounds. We sincerely hope that the achieved results will be useful for the development of and increase in LCB-based thermal insulation, contributing to the mitigation of climate change.

## 5. Conclusions

In the framework of the study, mold fungal resistance was evaluated for newly investigated loose-fill thermal insulation materials produced from wheat straw, corn stalk and water reed using different processing methods. Our results demonstrate that the mechanical milling of raw materials alone does not attenuate mold growth even with the addition of fungicide. The same mold growth (~75%) was detected also for steam-exploded pulp (SEP) without added fungicides. However, thermo-mechanical pulping (TMP) or SEP with fungicides significantly inhibited mold growth (35% and 40%, respectively). Moreover, the significant fungal resistance of the TMP and SEP substrates was evaluated not only against the five inoculated mold species but also from microflora of airborne. Therefore, SEP and TMP of all of the tested raw materials are the most suitable end products for thermal insulation applications. However, our laboratory-scale results should be further verified by long-term testing in a real construction project.

## Figures and Tables

**Figure 1 polymers-16-00562-f001:**
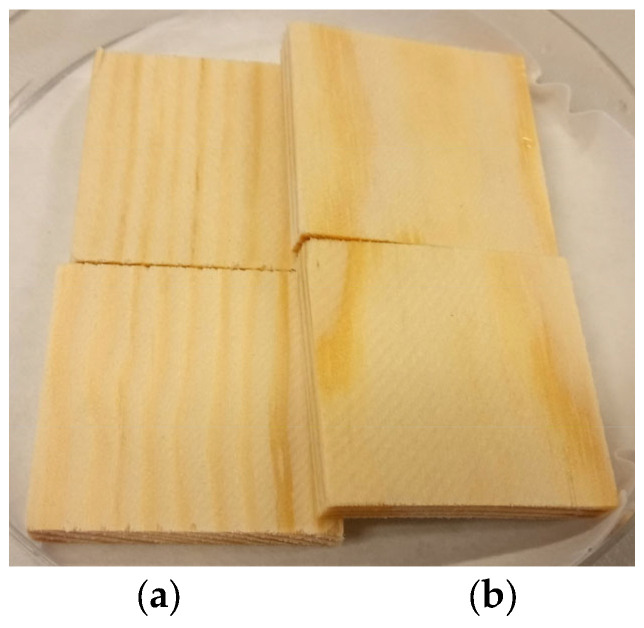
Radial (**a**) and tangential (**b**) surfaces of pine sapwood specimens.

**Figure 2 polymers-16-00562-f002:**
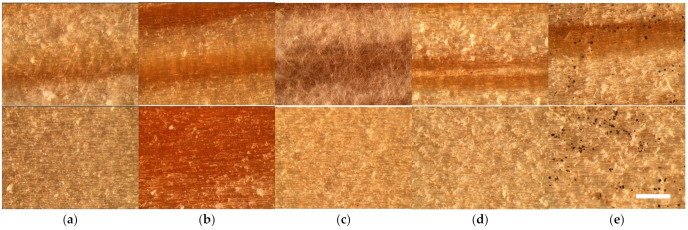
Microscopical surface view (25×, scale bar 1 mm) of mold fungal colonization on radial (**above**) and tangential (**bottom**) surfaces of pine sapwood specimens after 4 weeks incubation with (**a**) *Trichoderma viride*, (**b**) *Chaetomium globulosum*, (**c**) *Paecilomyces variotii*, (**d**) *Penicillium pinophilum* and (**e**) *Aspergillus niger*.

**Figure 3 polymers-16-00562-f003:**
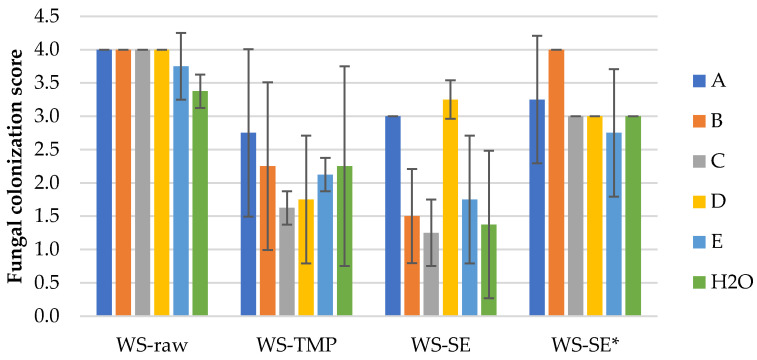
Mold fungal colonization (A–H2O, according to Table 1) of differently processed wheat straw samples. Error bars are one standard deviation of 4 specimens mean.

**Figure 4 polymers-16-00562-f004:**
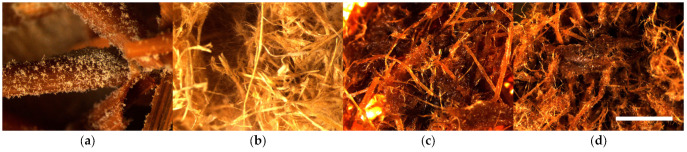
Microscopical surface view (10×, scale bar 3 mm) of mold fungal colonization on wheat straw samples after 4 weeks incubation: (**a**) WS-raw + A—*Trichoderma viride*, (**b**) WS-TMP + B—*Chaetomium globulosum*, (**c**) WS-SE + C—*Paecilomyces variotii* and (**d**) WS-SE* (H2O).

**Figure 5 polymers-16-00562-f005:**
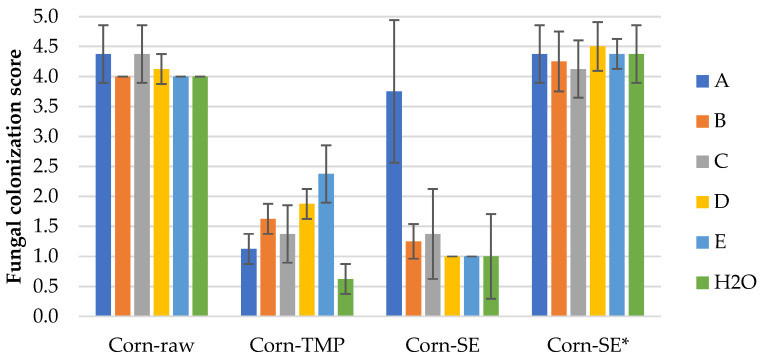
Mold fungal colonization (A–H2O, according to Table 1) on differently processed corn stalk samples. Error bars are one standard deviation of 4 specimens mean.

**Figure 6 polymers-16-00562-f006:**
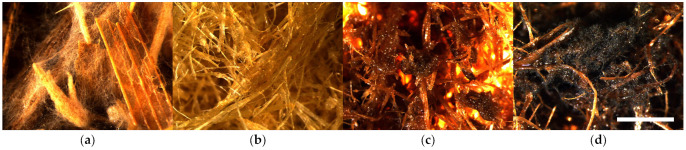
Microscopical surface view (10×, scale bar 3 mm) of mold fungal colonization on corn stalk samples after 4 weeks incubation: (**a**) Corn-raw + E—*Aspergillus niger*, (**b**) Corn-TMP (H2O), (**c**) Corn-SE + C—*Paecilomyces variotii* and (**d**) Corn-SE* + D—*Penicillium pinophilum*.

**Figure 7 polymers-16-00562-f007:**
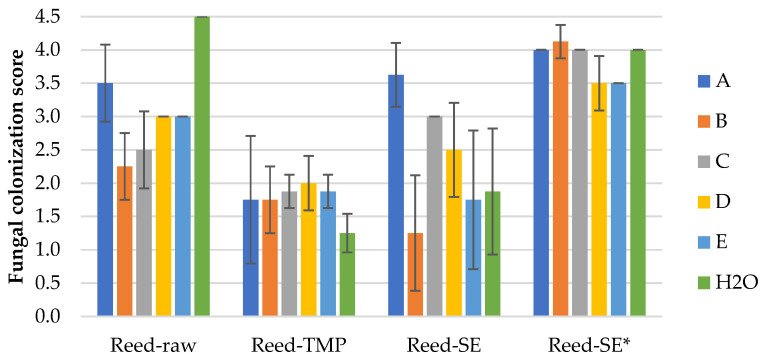
Mold fungal colonization (A–H2O, according to Table 1) of differently processed water reed samples. Error bars are one standard deviation of 4 specimens mean.

**Figure 8 polymers-16-00562-f008:**
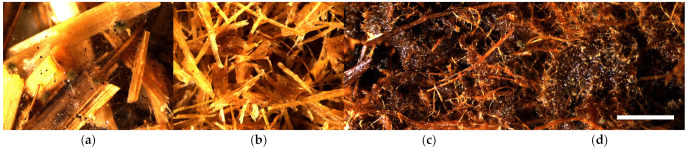
Microscopical surface view (10×, scale bar 3 mm) of mold fungal colonization on water reed samples after 4 weeks incubation: (**a**) Reed-raw + A—*Trichoderma viride*, (**b**) Reed-TMP (H2O), (**c**) Reed-SE + D—*Penicillium pinophilum* and (**d**) Reed-SE* + B—*Chaetomium globulosum*.

**Table 1 polymers-16-00562-t001:** Summary of mold fungi-exposed samples.

Fungi/ Sample	A *Trichoderma viride*	B *Chaetomium globulosum*	C *Paecilomyces variotii*	D *Penicillium pinophilum*	E *Aspergillus niger*	H2O
LCB ^1^-SEP	Steam-exploded pulp of each raw LCB					
LCB-SEP*	SE pulp of each raw LCB without fire retarder and fungicide					
LCB-TMP	Thermo-mechanical pulp of each raw LCB					
LCB-raw	Untreated chopped (sieve 20 mm) raw LCB					
Control	Untreated pine sapwood (30 × 30 × 5 mm)					

^1^ Each raw LCB material, e.g., WS (wheat straw), reed and corn.

**Table 2 polymers-16-00562-t002:** Mold fungal colonization scores of evaluated pine sapwood samples.

Surface	A	B	C	D	E	H2O
Radial	3	1	3	0	3	0
Tangential	0	1	0	0	3	0

## Data Availability

The raw data presented in this study are available on request from the corresponding author.

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
