# Peer review of "Mold Fungal Resistance of Loose-Fill Thermal Insulation Materials Based on Processed Wheat Straw, Corn Stalk and Reed"

_polymers, 2024, doi:10.3390/polym16040562_

Round 1
Reviewer 1 Report
Comments and Suggestions for Authors
The research of mold fungal resistance of loose-fill thermal insulation materials (based on Processed Wheat Straw, Corn Stalk and Reed) was presented in this paper. During the research the mold fungal resistance was evaluated for newly investigated loose-fill thermal insulation materials produced from wheat straw, corn stalk and water reed using different processing. It was noticed that in spite of fungicides addition, in average, heavy mold growth was detected on all raw materials covering. It has been shown that the highest resistance to internal and externally applied mold colonization among the tested source materials is demonstrated by Corn-TMP and Corn-SEP samples followed by Reed-TMP, wheat straw SEP and TMP and Reed-SEP.
The work is good described.
The referenceswas well chosen to the topic .
Some suggestion follows:
- it could be good to revise the keywords,
- it could be good to enhance the literature review in the Introduction,
- conclusions should be more specific,
- it could be good to Conclusion chapter rewrite, to well understanding by readers.
In my opinion after minor corrections, the work could be submitted for publication.
Author Response
Dear reviewer,
Thanks for your valuable review and suggestions to improve our manuscript!
Below, we are sending our response to your objections.
- it could be good to revise the keywords,
The keywords were revised.
- it could be good to enhance the literature review in the Introduction,
The introduction was enhanced by some extra-related works.
- conclusions should be more specific,
- it could be good to Conclusion chapter rewrite, to well understanding by readers.
Thank you for the suggestion. The Conclusion chapter was rewritten.
Reviewer 2 Report
Comments and Suggestions for Authors
This manuscript presented an interesting study about the mold fungal resistance of thermal insulation materials based on biomass. The work has potential. However, some points listed below need to be improved.
Abstract: please add more numerical results to the abstract.
Figure 1: I suggest add Figure 1 to the methodology section. In addition, what are the dimensions of the samples showed in Figure 1/
Figure 2, Figure 4, Figure 6, Figure 8: please add a scale bar in all these figures.
Supplemental material: please also add a scale bar in all figures presented in the supplemental material.
One general comment: In the discussion section the authors attributed some differences to the “soluble sugars” presented in the samples. However, the authors did not presented the composition (cellulose, hemicellulose, lignin,…) of the samples tested. Is it possible that differences on the sample composition may affect the fungal resistance results?
Author Response
Dear reviewer,
Thanks for your valuable review and suggestions to improve our manuscript!
Below, we are sending our response to your objections.
- Abstract: please add more numerical results to the abstract.
R: Thank you for the suggestion; however, the abstract includes the main numerical results already exceeding the allowed word amount (226/200) and in the case of adding some extra numerical results, it would be unacceptable by the journal requirements.
- Figure 1: I suggest add Figure 1 to the methodology section. In addition, what are the dimensions of the samples showed in Figure 1
R: Figure 1 was moved to the methodology section, where the dimensions of wood specimens were indicated as well.
- Figure 2, Figure 4, Figure 6, Figure 8: please add a scale bar in all these figures.
- Supplemental material: please also add a scale bar in all figures presented in the supplemental material.
R: Figures 2, 4, 6, 8 in the main manuscript and all Figures of the Supplementary Materials were modified by adding scale bars.
- One general comment: In the discussion section the authors attributed some differences to the “soluble sugars” presented in the samples. However, the authors did not presented the composition (cellulose, hemicellulose, lignin,…) of the samples tested. Is it possible that differences on the sample composition may affect the fungal resistance results?
R: The chemical composition of the study samples was outside of the study. However, the composition of some samples from this study was tested before and therefore attributed to this study for the explanation of mold fungi resistance. Because the negative effect of sugar content on mold growth was found by other authors mentioned in the Discussion section. This part was supplemented by the citation of some additional works approving the effect of different composition of wood substrates on fungal resistance.
Reviewer 3 Report
Comments and Suggestions for Authors
The topic is interesting but it is not clearly new. I have some reminders:
Figure 2. Add photo of control.
Figure 3, 5, 7. Write number of replicates
Figure 3. Why some of the standard deviations are so large? Is possible to exclude some data?
Figure 3, 5, 7. Explain letters (A, B, C, D, E, H20)
Add other literature sources about a treatment and use of straws materials in building engineering, The research studies are not complet.
Add at least a paragraph to the discussion or conclusions about what actually follows from your data. Do you recommend these materials for practical use? Which one of them seems to be the most suitable? Was it a good idea to use borate? It is a toxic substance, it is possible that soon it will be banned even for these purposes.
What are your plans for the future? Will you continue this study?....
I do not agree that „Together, the achieved fungal colonization scores after combined fungicide and pulping treatments are acceptable for the application of various renewable materials in future thermal insulation product development." All of your samples grew moldy and significantly so, even though they were treated differently.
Generally, discuss more your data with literature.
Comments on the Quality of English Language
Minor revision of English is need.
Author Response
Sear Reviewer,
Thanks for your valuable review and suggestions to improve our manuscript!
Below, we are sending our responses (R) to your objections.
-The topic is interesting but it is not clearly new.
R: The topic is not new, of course, however, it includes different unexplored processing’s of potential raw materials for production of biobased thermal insulation. In turn, the topic of fungal resistance of biobased thermal insulation materials is clearly missing.
- Figure 2. Add photo of control.
R: The photo of control in Figure 2 is not necessary since the sample did not grow molds and looks the same as the sample shown in Figure 2d. This is indicated in the text.
- Figure 3, 5, 7. Write number of replicates
R: The number of replicates was added.
- Figure 3. Why some of the standard deviations are so large? Is possible to exclude some data?
R: Four replicates were laboratory-tested per one sample of each material type. Therefore, there is no reason to exclude some data. The tested materials are loose-fill, not so homogenous, and therefore represent the biological variability of mold growth. The highest variability of the results demonstrates TMP and SEP samples, especially from wheat straw species. That indicates a possible influence of raw materials, uniform processing conditions, and even the spray-inoculation process of fungi spores.
- Figure 3, 5, 7. Explain letters (A, B, C, D, E, H20)
R: The indication of letters in Table 1 was added to all appropriate Figures.
- Add other literature sources about a treatment and use of straws materials in building engineering, The research studies are not complet.
R: Some extra sources were added in the Introduction about biomass pretreatment for the development of building materials.
- Add at least a paragraph to the discussion or conclusions about what actually follows from your data. Do you recommend these materials for practical use? Which one of them seems to be the most suitable? Was it a good idea to use borate? It is a toxic substance, it is possible that soon it will be banned even for these purposes.
R: The last paragraph of Discussion section includes three points about your questions. Because of environmental issues, the use of borates is still under question. However, it is still the main cheap additive in many commercial biobased insulation products, preventing them from both biological and fire threats.
- What are your plans for the future? Will you continue this study?....
R: This question was already addressed in the last paragraph of Discussion section.
- I do not agree that „Together, the achieved fungal colonization scores after combined fungicide and pulping treatments are acceptable for the application of various renewable materials in future thermal insulation product development." All of your samples grew moldy and significantly so, even though they were treated differently.
R: The research on mold fungal resistance of any biobased materials could be performed based on appropriate standards. All these standards require the appropriate constant laboratory conditions to enable the perfect fungi growth. That allows the testing of the fungus growth intensity on various materials depending on many factors, such as raw species, material pretreatment, and processing. This was done in our study approving the significant effect of TMP and SEP processing of selected raw materials combined with fungicide on effective resistance against mold fungi. The results of the study are very promising for the application of tested renewable materials in future development of thermal insulation products. Furthermore, in real life, the weather conditions in which the building materials are located are not constant; therefore, of course, the developed materials should be tested in real construction for a longer time to completely verify the results of the study. And we are going to do that in further study.
- Generally, discuss more your data with literature.
R: Thank you for the suggestion. Our results were discussed more extensively with the literature in the Discussion section.
Round 2
Reviewer 3 Report
Comments and Suggestions for Authors
I feel that new version of manuscript is suitable fot publishing.